# From emergence to endemicity of highly pathogenic H5 avian influenza viruses in Taiwan

Yao-Tsun Li [1] ✉, Hui-Ying Ko[2], Joseph Hughes [3], Ming-Tsan Liu[4], Yi-Ling Lin [2,5], Katie Hampson [1] & Kirstyn Brunker [1,3] ✉

A/goose/Guangdong/1/96-like (GsGd) highly pathogenic avian influenza (HPAI) H5 viruses cause severe outbreaks in poultry when introduced. Since emergence in 1996, control measures in most countries have suppressed local GsGd transmission following introductions, making persistent transmission in domestic birds rare. However, geographical expansion of clade 2.3.4.4 sublineages has raised concern about establishment of endemic circulation, while mechanistic drivers leading to endemicity remain unknown. We reconstructed the evolutionary history of GsGd sublineage, clade 2.3.4.4c, in Taiwan using a time-heterogeneous rate phylogeographic model. During Taiwan's initial epidemic wave (January 2015 - August 2016), we inferred that localised outbreaks had multiple origins from rapid spread between counties/cities nationwide. Subsequently, outbreaks predominantly originated from a single county, Yunlin, where persistent transmission harbours the trunk viruses of the sublineage. Endemic hotspots determined by phylogeographic reconstruction largely predicted the locations of re-emerging outbreaks in Yunlin. The transition to endemicity involved a shift to chicken-dominant circulation, following the initial bidirectional spread between chicken and domestic waterfowl. Our results suggest that following their emergence in Taiwan, source-sink dynamics from a single county have maintained GsGd endemicity up until 2023, pointing to where control efforts should be targeted to eliminate the disease.

Preventing emerging zoonotic viruses from establishing endemic circulation is essential to public health, considering the great health burden caused by endemic zoonoses globally[1]. An example of particular concern is the A/Goose/Guangdong/96-like (GsGd) H5 viruses, a lineage of highly pathogenic avian influenza (HPAI) first identified in southern China. From 2003 to 2006, HPAI GsGd viruses spread globally, infecting poultry in countries across Asia and Europe[2]. Most countries have subsequently eliminated HPAI GsGd viruses[3,4], but since

2010, a few countries, including Bangladesh, China, India, Indonesia, and Vietnam, have been classified as endemic, due to continued circulation in poultry and sporadic human cases[5]. Although there has been extensive characterisation of the genetic origins and the migration routes of HPAI GsGd viruses[6–8], the mechanisms facilitating persistence and the transition to endemicity remain unknown.

Since emergence, GsGd viruses have undergone diversification by accumulating mutations in their surface protein hemagglutinin (HA),

[1]School of Biodiversity, One Health & Veterinary Medicine, University of Glasgow, Glasgow, UK. [2]Institute of Biomedical Sciences, Academia Sinica, Taipei, Taiwan. [3]MRC-University of Glasgow Centre for Virus Research, Glasgow, UK. [4]Center for Diagnostics and Vaccine Development, Centers for Disease Control, Ministry of Health and Welfare, Taipei, Taiwan. [5]Biomedical Translation Research Center, Academia Sinica, Taipei, Taiwan. ✉e-mail: Yao-Tsun.Li@glasgow.ac.uk; Kirstyn.Brunker@glasgow.ac.uk

resulting in multiple antigenically distinct sublineages, termed clades[2,3]. These genetic changes can alter host preferences, determining the virus' reservoir, dissemination and persistence[9]. GsGd viruses of the clade 2.3.4.4, including clade 2.3.4.4a-2.3.4.4h, are distinguished from previous GsGd viruses by various neuraminidase (NA) subtypes (N2, N5, N6 and N8) besides N1, and have rapidly increased in the global population since 2014[10,11]. These "H5Nx" viruses have caused poultry and wild animal outbreaks in previously unaffected regions, such as the Americas[10,12], and have also demonstrated unexpected transmission from Europe back to China[13]. Moreover, the viruses have led to mammal-to-mammal transmission in minks and cows[14,15], and at a much larger scale among marine mammals[16]. The increasingly sustained circulation in areas previously thought to be sinks raises concerns about expanding the establishment of endemic circulation[17]. Studying the development of long-term virus circulation could reveal insights into approaches to eliminate viruses from areas currently affected by GsGd clade 2.3.4.4.

The clade 2.3.4.4c virus, one of the H5Nx sublineages, caused severe outbreaks in Korean poultry farms in early 2014[10,18]. Following migratory bird flyways, the virus spread to Japan, the USA and Taiwan by the end of that year[19–21]. In 2015, Taiwan experienced a devastating epidemic in domestic birds and culled over 5 million birds to curb localised outbreaks[22]. This marked the first significant transmission of HPAI GsGd viruses in the country[23,24]. While clade 2.3.4.4c viruses were eliminated in other countries before 2017[11,21], the virus continued to circulate as the H5N2 subtype in Taiwan until at least 2019 from the original H5N8 subtype[21,25]. With multiple genotypes generated by reassortment with local low pathogenic avian influenza (LPAI) viruses[21,24], Huang et al. recognised the circulation of the HPAI clade

2.3.4.4c virus in Taiwan as endemic[22], and the circulating viruses were all linked to the same introduction event occurring before 2015[21,22]. How this clade dispersed in Taiwan and the factors underlying its endemic establishment are not understood. In addition, the initial 2015 epidemic wave in Taiwan was characterised by widespread transmission in domestic waterfowl[26]. As there is no evidence of HPAI circulating in resident wild birds in Taiwan, the population dynamics of the virus in different poultry species is unclear.

In this study, we quantified the persistence of GsGd outbreaks globally using virus HA genes and identified the source of the endemic clade that was established in Taiwan within this global context. We then curated the HA genes of the clade 2.3.4.4c virus in Taiwan, isolated from January 2015 to March 2019, representing ~20% of reported outbreaks in poultry that occurred within the country. Using this data set, we performed time-heterogeneous phylogeographic analyses to reconstruct the geographical and ecological dispersal dynamics of the clade as it transitioned from emergence to endemicity. Specifically, we elucidate the factors facilitating viral spread among counties/cities in Taiwan following the first epidemic wave in 2015. Lastly, we investigated the viral spread between different host groups.

## Results
### Global persistence of GsGd
To better understand local epidemics following GsGd introductions on a global scale, we identified descendent sublineages caused by introductory events (see the "Methods" section). We found about 400 sublineages by reconstructing dispersal history from a uniformly downsampled dataset using the H5 genes of GsGd viruses up to September 2022 (Fig. 1). These GsGd sublineages circulated across

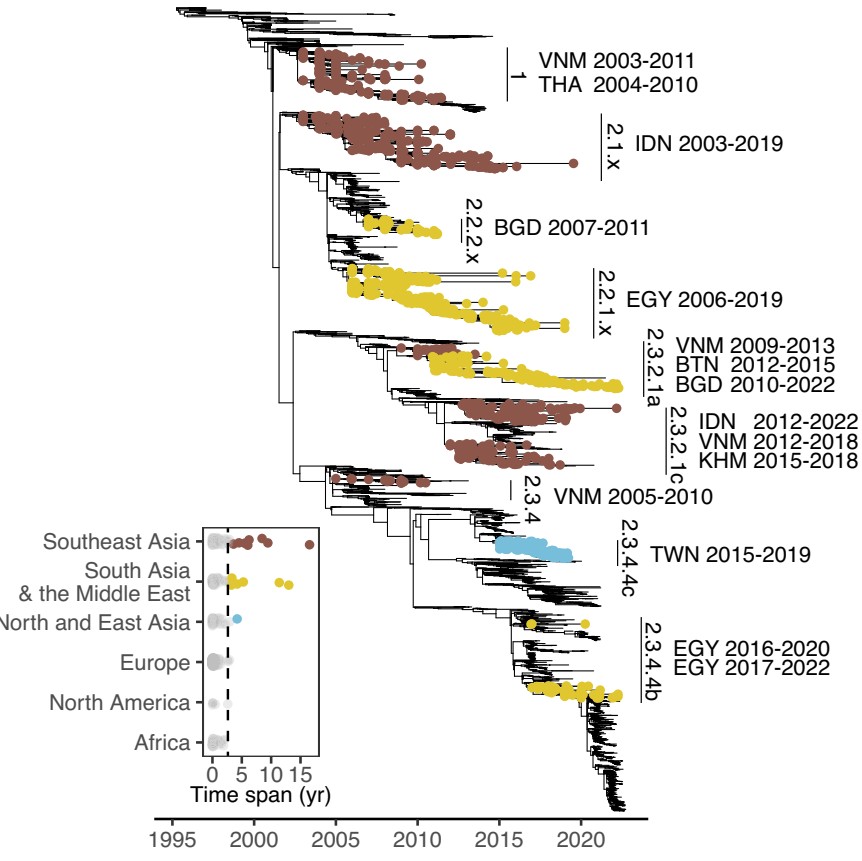

**Fig. 1 | The duration of GsGd outbreak lineages.** The time-scaled phylogenetic tree inferred with HA genes highlights 15 lineages, which persistently circulated for at least 3 years. These are labelled with H5 nomenclature classification, country of isolation and duration. The inset shows the lineage duration classified by geographical area, with persistent sublineages highlighted as tips on the phylogeny. The dashed line denotes 95% quantile. VNM Vietnam, THA Thailand, IDN Indonesia, BGD Bangladesh, EGY Egypt, BTN Bhutan, KHM Cambodia, TWN Taiwan.

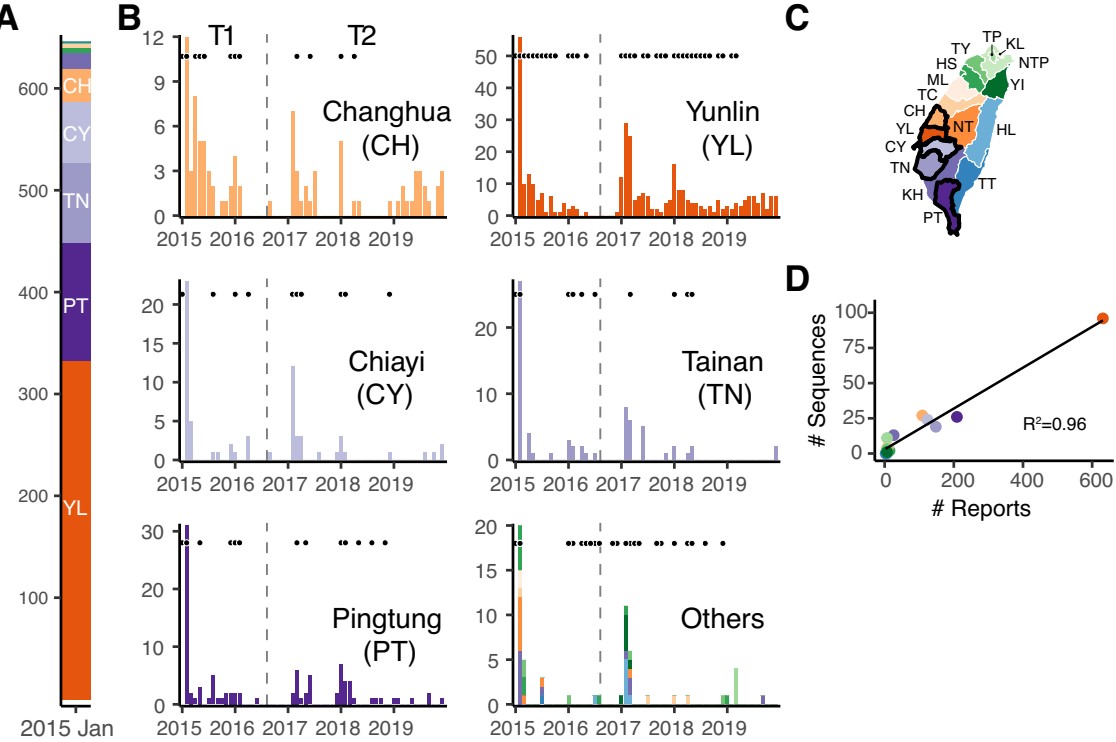

**Fig. 2 | The emergence of HPAI GsGd clade 2.3.4.4c in Taiwan and subsequent circulation from 2015 to 2019. A** and **B** The number of reports of infected poultry in farms or abandoned dead poultry. The January 2015 numbers are pooled in panel **A**, while the monthly numbers of reports are presented by location in panel **B**. Black dots above the bars indicate the availability of genomic data in the corresponding months. The dashed vertical lines indicate the start of September 2016, the transition from the emergence (T1) to the endemic phase (T2). **C** The map highlights the five counties/cities with the most reports. **D** The correlation between reported outbreaks and available sequences for each of the 14 counties/cities. The points are coloured using the same colour scheme as the map and the other panels. NTP New Taipei, TP Taipei, KL Keelung, TY Taoyuan, HS Hsinchu, YI Yilan, ML Miaoli, TC Taichung, CH Changhua, NT Nantou, YL Yunlin, CY Chiayi, TN Tainan, KH Kaohsiung, PT Pingtung, HL Hualien, TT Taitung.

78 countries, but their circulation was generally short-lived, with an average duration of 0.6 years based on the date of sample collection. Closer examination of sublineages that persisted for more than three years (based on a 95% quantile of 2.64), revealed 15 persistent sublineages in Southeast Asia, South Asia and the Middle East, including the clade 2.3.4.4c virus in Taiwan (Fig. 1). Geographical reconstruction informed by outbreak reports supported this finding that most introductions had limited onward transmission (on average, 0.61 years) and identified the same countries with prolonged viral circulation.

## The emergence and spread of clade 2.3.4.4c in Taiwan

In January 2015, over 600 outbreaks of clade 2.3.4.4c were reported in Taiwan across 10 counties/cities (Fig. 2A). Subsequently, reported HPAI outbreaks in Taiwan declined and became sporadic in previously affected areas, with only three cases reported during July–August 2016, before transmission resumed in winter 2016–2017 (Fig. 2B). To understand the spread of the clade 2.3.4.4c virus in Taiwan, we curated a genetic data set of viral HA sequences (*n* = 252), including subtypes of H5N2 (73%), H5N3 (2%) and H5N8 (25%). The sequences are distributed over time from January 2015 to March 2019, and their sample sizes per county/city correlate well with the number of reported outbreaks (Fig. 2D).

To understand the dispersal dynamics of the 2.3.4.4c clade in Taiwan, we reconstructed the viral spread using continuous phylogeographical approaches, which inferred the spatial locations of nodes. The Bayesian MCMC sampling process used priors informed by surveillance data to specify the locations of viruses accommodating uncertainty (see the "Methods" section). The results, stratified by year, indicate that in 2015 there was intensive virus dispersal within and

between western counties/cities (CH, YL, CY, and TN), as well as more limited dispersal in southern counties/cities (KH and PT) (Fig. 3). Long-distance dispersal events mostly originated from the main cluster in the west, and the northward dispersal events did not appear to seed sustained outbreaks. Since 2016, the number of dispersal events has significantly decreased, and since 2017, Yunlin County (YL) has become the primary source of both short- and long-distance spread (Fig. 3). The pattern is further illustrated by discretizing the inferred locations, with diffusion within YL dominating in all directions (Supplementary Fig 1).

Use of explicit phylogeographic methods can reduce sampling biases[27,28]. Here, the locations of samples from slaughterhouses or rendering factories introduced inaccuracies into our inference. This is because these locations were unlikely to be the origins of viral spread, particularly those from the northern cities (e.g. TP and NTP) where agricultural activity is minimal (Supplementary Fig. 2). To reconstruct the dispersal history of the virus, we therefore used discrete diffusion methods to reassign the locations of slaughterhouse samples. Specifically, we employed priors informed by the surveillance data, which represented possible sources of animals transported to slaughterhouses or rendering factories by uncertain discrete state assignment (Supplementary Fig. 3B). Additionally, we used a heterogeneous rate model to identify changes in dispersal patterns between two epidemiological phases defined from the surveillance data (Fig. 2): from January 2015 to August 2016 (T1) and from September 2016 onwards (T2).

The statistically supported routes that we inferred indicate a shift in dispersal from outbreaks having multiple sources to predominantly originating from a single source (Fig. 4 and Supplementary Fig. 4A) and

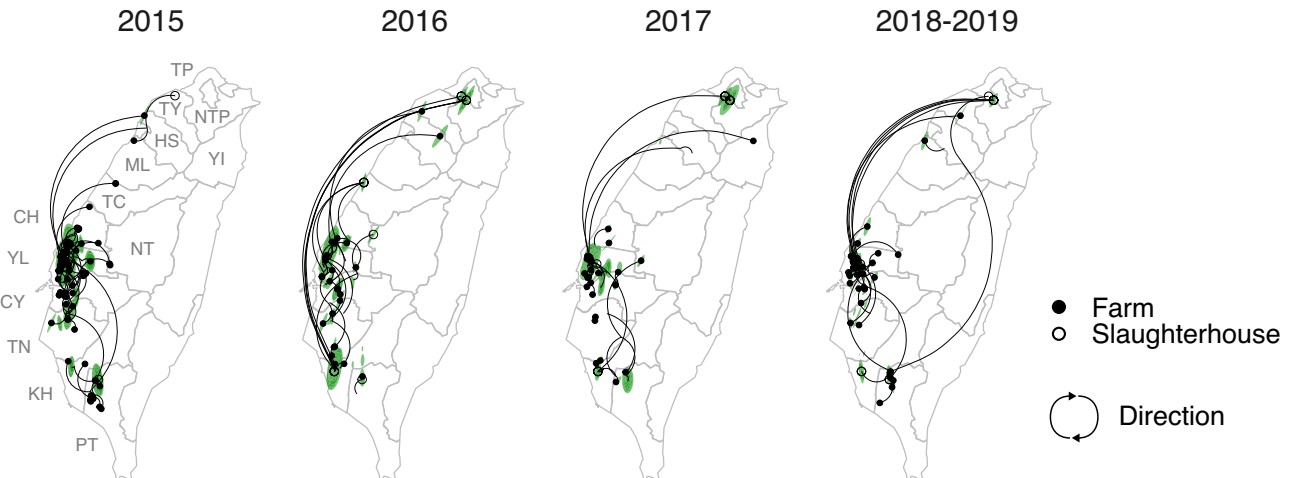

**Fig. 3 | Reconstruction of the dispersal of clade 2.3.4.4c in Taiwan, 2015–2019.** The points on the maps indicate the location of the virus samples, while the coloured areas represent the 80% HPD (highest posterior density) of the nodes inferred by the continuous phylogeography. Curves represent the branches of the maximum clade credibility (MCC) tree and are categorised into four periods based on the timing of parental nodes. Dispersal direction (clockwise) of the viral lineage is indicated by the curvature.

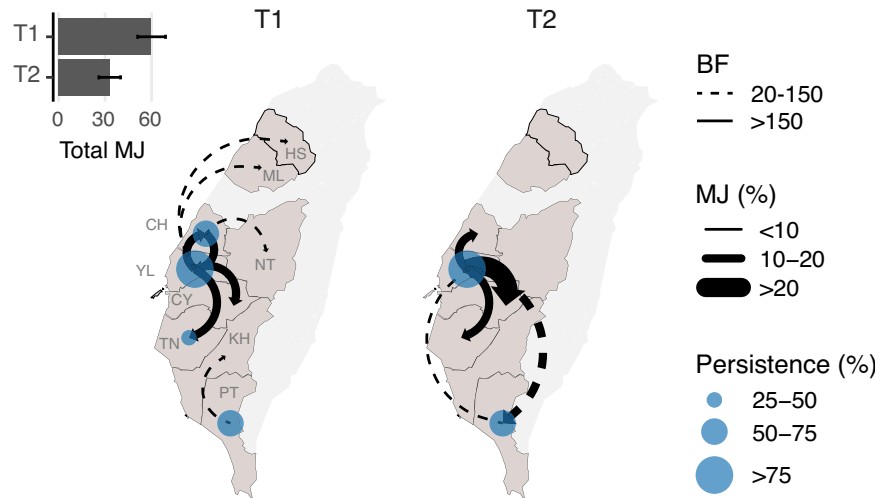

**Fig. 4 | Schematic illustration of the main diffusion routes of the Taiwan clade 2.3.4.4c virus during the first epidemic wave and subsequent transition to endemicity.** The statistically supported directions between counties/cities (Bayes factor > 20) inferred by the discrete phylogenetic analyses are indicated as curves with arrows. The different line types reflect the strength of support (BF). The thickness of the curve reflects the frequency of transition events, presented as the proportion of total Markov jumps in the time period (T1 or T2). Total Markov jumps are indicated on the upper left inset. The size of the blue circles reflects the degree of persistence, calculated by unifying phylogeny branches with both nodes inferred as the same geographical area divided by the T1/T2 time period. Only areas where viruses circulated for more than 25% of the period are labelled. T1, January 2015 to August 2016; T2, September 2016 to March 2019.

are generally consistent with inference from the continuous phylogeographic analyses. During the emergence phase (T1), substantial transition events were detected between two western counties (YL and CH), in addition to dispersal from YL to nearby counties, with less frequent transitions identified from CH and PT to neighbouring counties/cities. In contrast, during the transition to endemicity (T2), all supported routes originated from YL, except for a less frequent path from PT. To account for local transmission, we quantified the maximum time interval assembled by branches with the same inferred state (Fig. 4 and Supplementary Fig. 5A). YL appears to be the only site where viruses persisted for more than 75% of both time periods (T1 and T2), suggesting its potential as a source of virus spread. Furthermore, when considering the viruses leading to the most recent isolates, CH harboured the "trunk" of the clade in 2015, but since 2016, YL has

consistently been the major site harbouring viruses leading to the most distant progeny (Supplementary Fig 6A). These results suggest that YL has become the most prominent origin for nationwide dispersal following the initial epidemic wave. In contrast, circulation in other locations, particularly CH, has been largely suppressed, preventing the spread to other counties/cities.

To investigate the impact of sampling, we created two subsampled data sets by downsampling genetic sequences from county YL, resulting in YL no longer being the most frequently sampled site in each dataset. The subsampled data showed a shift in the origins of viral diffusion to CH or TN during the emergence phase (T1), but the inferred routes during the endemic phase (T2) were almost identical to those obtained using the full dataset (Supplementary Fig. 4B and C). Similarly, persistence quantified from subsampling shows that CH

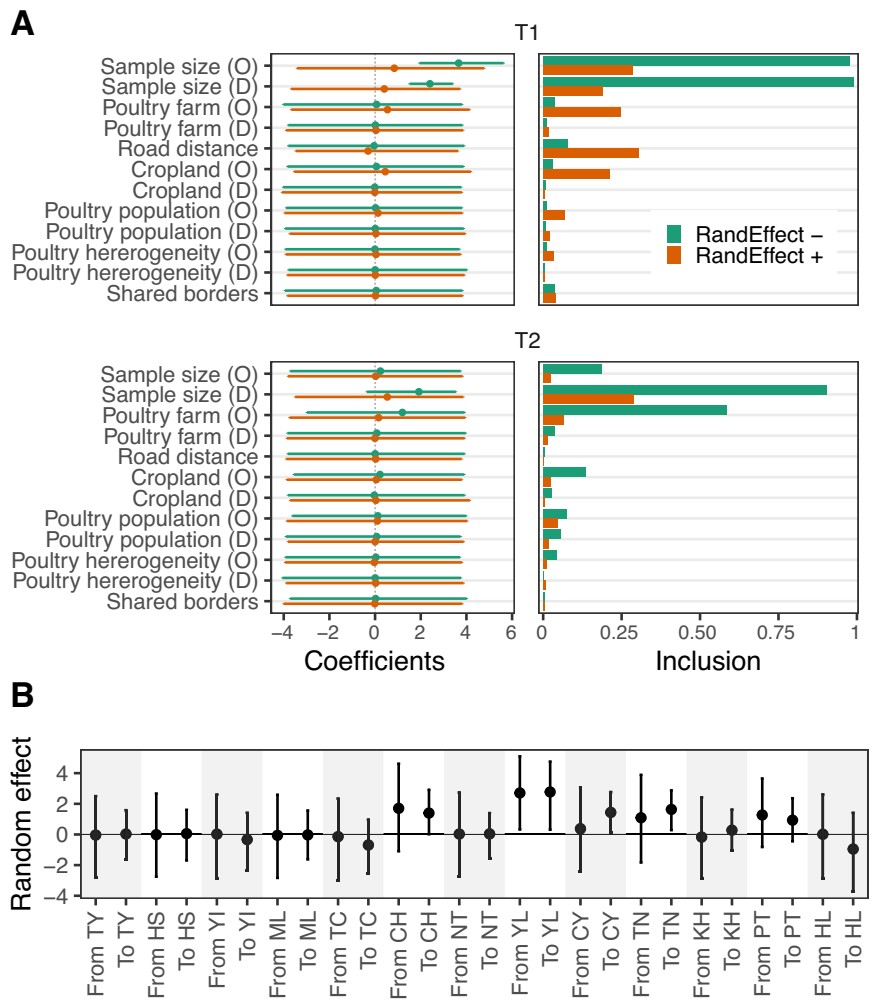

**Fig. 5 | Potential predictors of dispersal of clade 2.3.4.4c virus between counties/cities in Taiwan. A** The conditional effect sizes quantified by coefficients (left) and the inclusion probabilities of predictors (right) estimated by the time-heterogeneous phylogenetic generalised linear model (GLM). Results of the models with and without random effect variables are shown. The predictor names are denoted by O in parentheses for origin and D for destination. **B** Location-specific random effects in the GLM model. The effect sizes are in log space and are presented as mean with 95% HPD interval.

replaced YL as the location with the most persistent circulation during the initial epidemic wave (T1). However, the estimates during the endemic phase (T2) were largely unaffected by the targeted sub-sampling (Supplementary Fig. 5B and C).

To understand how the surveillance data may have biased our inference, we also performed the same phylogeographic approaches based on the original collection county/city, with unknown samples (sample size; T1:10, T2:1) assigned an uninformative prior containing all collection locations. This inference, which is independent of outbreak records, reveals a more widespread distribution of dispersal origins during the initial epidemic wave (T1). In addition to PT, more southern locations (i.e. TN and KH) were also identified as dispersal origins (Supplementary Fig. 4D). During the endemic phase (T2), the pattern is consistent with the results obtained from considering the outbreak records. Notable differences were found in routes involving locations where samples were mainly from slaughterhouses or rendering factories (i.e. NTP and TP, Supplementary Fig. 4D). Estimates of persistence based on record-independent priors (Supplementary Fig 5D) and all trunk probability conditions tested (Supplementary Fig. 6B–D) indicate similar results to the full data set. Taken together, these findings suggest that our observation of nationwide geographical spread is robust to both sampling and state assignment.

## Location-specific factors drive the risk of viral dispersal

To identify the risk factors that facilitated the spread of clade 2.3.4.4c in Taiwan, we implemented a Generalised Linear Model (GLM) model accommodated by the time-heterogeneous discrete phylogeographic method. This allowed for estimates of predictor effect size and inclusion probability by time period. The results show that sample size is positively and significantly correlated with diffusion during the initial epidemic wave (T1, Fig. 5A, RandEffect-). The number of poultry farms has a prominent inclusion probability during the endemic phase (T2), but this was not statistically significant. To account for location-specific factors, the GLM model was adjusted by adding random effects for each site, in addition to the existing predictors. The new model reduced the effects of sample size, and no further predictors were identified (Fig. 5A, RandEffect+). Interestingly, the result shows that dispersal routes to high-incidence counties/cities (CH, YL, CY and TN), and the direction from YL, have significant positive effects (Fig. 5B). To avoid overparameterization, we conducted GLM estimations with only one of the predictors that had higher inclusion probabilities and sample size. Consistent with the full model, the reduced models that include predictors for either poultry farm, road distance, or cropland show no significant effects for these factors (Supplementary Fig. 7). Furthermore, the random effects that specify directions to YL, CY, or TN are significant in all three reduced models, while

the effect of the direction from YL is significant in one reduced model. The results remained consistent when the full model was examined in a time-homogeneous manner (Supplementary Fig. 8). These findings suggest that location-specific factors are more likely to explain the spatial spread of the virus in Taiwan than shared agricultural risk factors.

## Most YL outbreaks re-emerge from endemic hotspots

The phylogeographic results suggest a pivotal role of YL county in driving outbreaks across the country. However, the determinants that distinguish YL country from other locations remain unclear. To investigate this, we evaluated re-emerging HPAI outbreaks in the persistent foci during the endemic phase (T2). Based on the practice of establishing surveillance zones after case identification[22,29], hotspots were defined by overlapping buffer areas within 1 km of node locations inferred by the continuous phylogeographic methods, or within the same distance of the node plus outbreak locations (Fig. 6A). When mapping new outbreaks in YL from 2019 to 2022, over 50% of these occurred within the phylogenetically-defined hotspots. Moreover, over 70% of the outbreaks were found in the hotspots defined by the phylogenies supplemented with contemporary outbreak sites (Fig. 6B). The areas defined by both methods account for <20% of the area of YL (Supplementary Fig. 9). When the buffer radius was set to 0.5 km, over 30% and 60% of the new outbreaks were still identified within the hotspots defined by the two methods, accounting for less than 10% of the county's area (Supplementary Fig. 9). When the approach is applied to four other high-incidence areas in Taiwan (Fig. 2B), YL shows significantly higher percentages of recurrence in the hotspots defined by the two buffer distances (Fig. 6C). These findings suggest that the inability to interrupt transmission within the county has led to the development of an endemic focus of persistent viral circulation in YL. Note that the

re-emerging outbreaks here may include HPAI of different H5 lineages, such as clade 2.3.4.4c reassortants, without available genomic data for further investigation.

## Development of chicken-dominant circulation

The initial 2015 widespread transmission in domestic waterfowl corresponded to a higher proportion of outbreaks and viral samples attributed to ducks or geese compared to chickens in 2015 (Fig. 7B and Supplementary Fig. 3A). To assess transmission between poultry species, we applied similar phylogeographic methods to the same genetic data, classified into Anseriformes (duck and goose) or Galliformes (chicken, turkey and quail) species. The inferred diffusion indicates that transmission between the two host groups was bidirectional during the initial epidemic wave (T1), with 80% of transmission attributed to spread from waterfowl, while both routes were statistically supported (Fig. 7A). During the endemic phase (T2), the transmission shifted to gallinaceous birds, while the transmission route from waterfowl was not supported (Fig. 7A). This finding was further confirmed by an NA data set containing cognate N2 genes of Taiwan clade 2.3.4.4c viruses (Supplementary Fig. 10). The trunk analyses show that the terrestrial poultry has maintained the gene source of the lineage since its introduction (Fig. 7C). These results suggest that terrestrial poultry, mainly chickens, served as reservoir for the clade 2.3.4.4c virus during the endemic phase (T2), following the first epidemic wave during which inter-host group transmission dominated and more likely originated from domestic waterfowl.

By detecting signals of positive selection, we assessed the evidence for host adaptation. The dN/dS (w) estimate of HA (H5) genes in Taiwan clade 2.3.4.4c viruses is similar to that of HA genes in a North American H5N2 lineage introduced to the Taiwanese chicken population as early as 2003[21,30], while the estimate of NA (N2) genes in the clade 2.3.4.4c viruses is higher than that of the North American lineage (Supplementary Table 1). In addition, there are less than five positively selected sites detected in both HA and NA of clade 2.3.4.4c in Taiwan, which is comparable to the results found in the North American H5N2 lineage. Note that all of the substitutions occurring at these sites were not fixed in the samples collected in 2019 (Supplementary Table 1). North American H5N2 has been strictly isolated in terrestrial poultry before 2000[21,30], illustrating long-term circulation in Galliformes hosts. These results thus do not support selection pressure from host adaptation during the transition to endemicity and establishment of chicken-dominant circulation.

## Discussion

The circulation of clade 2.3.4.4c in Taiwan presents an opportunity to study how the GsGd lineage established endemic transmission following its emergence. By comprehensively characterising the dynamics of clade 2.3.4.4c in Taiwan, this study focuses on viral dispersal and mechanisms driving endemic transmission after the initial epidemic wave. While the introduction and the genetic origins of the virus have been reported previously[21,24], the dispersal dynamics and mechanisms driving persistence were previously unknown. Our time-heterogeneous phylogeographic analyses show that nearly all viral diffusion after mid-2016 originated from a single county, in contrast to the interconnected and multi-origin diffusion between counties/cities during emergence in 2015 (Figs. 3 and 4). These endemic hotspots in Taiwan, where the disease was persistent, continued to seed outbreaks in other agricultural areas, while new outbreaks were mostly under control. Therefore, our geographical reconstruction provides insights into strategic control measures to eliminate GsGd viruses in Taiwan, emphasising the need to focus on the endemic hotspots. Note that the clade 2.3.4.4c virus was still being identified in 2023[25], although viral genetic data isolated after 2019 was unavailable. Release of new clade 2.3.4.4c sequences would allow further investigation.

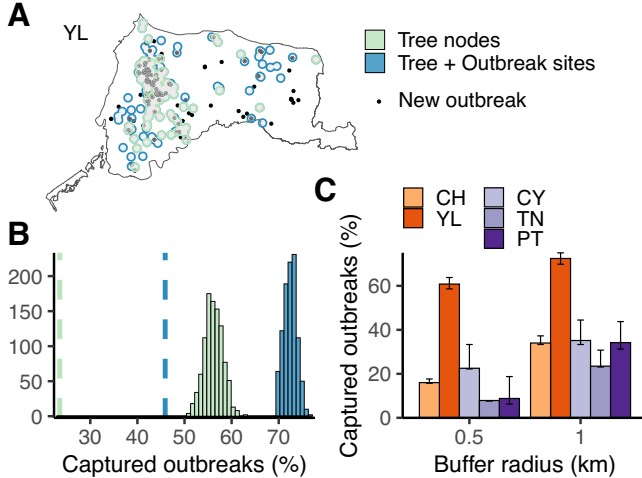

**Fig. 6 | Evaluating the spatial distribution of re-emergent H5 HPAI outbreaks in Yunlin county (YL). A** An example from a randomly selected posterior tree. The black points indicate reported outbreak sites that occurred between 2019 and 2022, after the latest available genetic data. The green/blue areas created by merged buffers indicate the inferred hotspots. **B** Distributions of captured new outbreaks by inferred epidemic hotspots based on 1 km-buffers with centres of nodes only (green) or nodes plus outbreak sites (blue). The vertical dashed lines indicate the mean of randomly generated circulating areas with identical buffer numbers as tree nodes (green) or tree nodes plus outbreak sites (blue). **C** Comparing the re-emergence of outbreaks within inferred hotspots in different locations. Proportions of captured outbreaks were calculated with buffers of two radius distances. The bars show the mean values based on 1000 posterior trees. Error bars indicate 95% credible intervals.

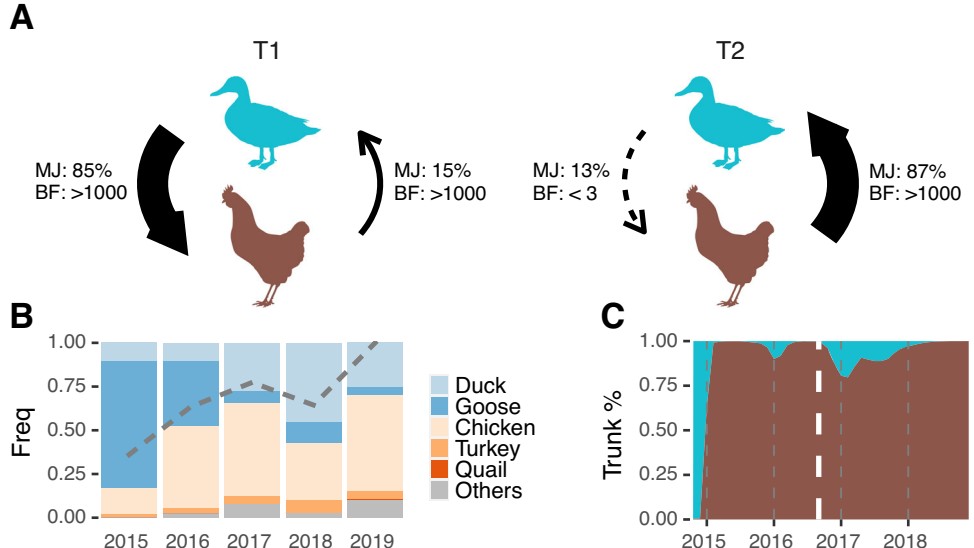

**Fig. 7 | Transmission shifted from predominantly domestic waterfowl to chicken during the transition from epidemic to endemic circulation of 2.3.4.4c in Taiwan. A** Shift in diffusion patterns between Anseriformes (duck and goose) and Galliformes (chicken, turkey and quail). Frequencies of transition events (Markov jump, MJ) and statistical support (Bayes factor, BF) were inferred using the discrete phylogenetic method. **B** The host distribution over time is based on the surveillance data. The grey dashed line indicates the proportions of Galliformes in the genetic data. **C** Inferred host categories, i.e. Anseriformes or Galliformes species, of the trunk of clade 2.3.4.4c. The white dashed line divides T1 and T2, with areas coloured using the same scheme as panel (**A**).

Previous studies using phylogenetic GLMs have identified the poultry trade[31], geographic distance[32] and the wild bird migration network[13] as predictors of GsGd viral dispersal. These studies have investigated transmission patterns globally as well as single epidemic waves, but have not examined the risk of establishment of endemic transmission. Our GLMs did not identify any significant agricultural predictors for GsGd viruses in Taiwan. Instead, our analyses highlighted location-specific factors that were not explained by the tested predictors. These results indirectly support the nationwide spread from a specific endemic county, with other risk factors playing only a minor role. The ongoing HPAI outbreaks occurring during 2023-2024 in Taiwan were mostly from Yunlin (dashboard in ref. 25, accessed 10 September 2024), supporting the proposed transmission pattern. It is possible that overparameterization due to sample size could not be excluded, although different models were extensively explored (Supplementary Figs. 7 and 8). Of all the predictors tested in the GLMs, the number of poultry farms had the highest inclusion probability (Fig. 5 and Supplementary Figs. 7 and 8), a finding corroborated by a spatial case-control study focusing on clustering of H5 viruses in Taiwan from 2015 to 2017[29].

Lineage-specific host preferences can be targeted to prevent viral spread[33]. Control measures for HPAI have targeted domestic waterfowl due to their biological similarity to wild waterfowl, which are capable of long-distance dissemination[8,34]. In addition, infected waterfowl exhibit mild clinical manifestations compared with Galliformes species like chickens, increasing the risk of silent transmission and co-infection with multiple strains[2]. Our diffusion analyses indicate that GsGd transmission in Taiwan shifted from bidirectionality, with mixing between domestic waterfowl (Anseriformes species) and terrestrial species (Galliformes species), to a predominantly chicken-origin circulation (Fig. 7). The greater transitions from waterfowl to terrestrial species during the first epidemic wave, aligned with devastating outbreaks in geese[22,26], that did not lead to sustained descendant lineages (Fig. 7C). Based on the selection analyses of the surface proteins, it is unlikely that the change in dispersal patterns between these epidemiological phases reflects virus adaptation from waterfowl to terrestrial birds (Supplementary Table 1). In addition, there were no notable molecular signatures of adaptation detected in the GsGd viruses in

Taiwan, compared with the North American H5N2 viruses in Taiwan, where N2 genes have 20 amino acid deletions indicating adaptation to terrestrial poultry[30]. Instead, assuming sampling strategies did not change dramatically during 2015-2019, the results suggest that unknown factors in the chicken farming networks supported endemic circulation of GsGd in Taiwan, while viral lineages circulating in goose and duck populations were mostly eliminated. During the transition to endemicity, more than half of the Anseriformes samples were either isolated or inferred to be from Yunlin county (Supplementary Fig. 3), indicating that the diffusion pattern in host ecology was not confounded by the spatial pattern described earlier.

Although these data have provided invaluable information on H5 HPAI circulation in Taiwan, it is unlikely that the surveillance data from the Taiwanese agricultural authorities identified all of the GsGd or clade 2.3.4.4c outbreaks. This limitation is implied by the results of the discrete diffusion analysis that relied solely on genetic sequences without outbreak records (Supplementary Fig. 4D). The analysis exclusively identified statistically supported transmission routes originating from locations in southern Taiwan, particularly city KH, during the first epidemic phase. It was also noted that some viral samples do not precisely correspond to inferred locations (Fig. 2B and see the "Methods" section). Our reconstruction schemes demonstrate that state assignment strategies have little effect on all tested characteristics during the endemic phase, T2 (Supplementary Figs. 4–6). There may also be an issue with the lack of explicit viral classification for each entry in the surveillance data, leading to inaccurate priors for the locations of the 2.3.4.4c virus. However, there is no evidence of the continued circulation of other GsGd lineages, including clade 2.3.4.4b, in Taiwan during 2015–2019, according to available sequences (see the "Methods" section) and outbreak reporting[25]. Most of the historical reconstruction in this study was inferred with HA; however, the effect of NA and other internal genes on virus transmissibility and host adaptation was unclear. The lack of samples from resident wild birds also limits our understanding of the potential role these birds play in the nationwide spread of the virus.

Our findings indicate that in Taiwan, HPAI control should prioritise the source region, specifically by strengthening measures to interrupt transmission and improve biosecurity in poultry farms in

Yunlin county. Meanwhile, epidemics in other regions could be readily suppressed. On a global scale, attention should be paid to countries newly affected by GsGd viruses after 2020[12,17], as well as countries where viruses were previously seeded (Fig. 1). Effective interventions can be implemented through continued local and global surveillance efforts to discern dispersal patterns supported by genomic data.

## Methods

### Sequence data preparation
H5 hemagglutinin (HA) sequences of avian influenza A viruses were downloaded from NCBI (Influenza Virus Resource, https://www.ncbi.nlm.nih.gov/genomes/FLU/)[35] and GISAID (EpiFlu, https://www.gisaid.org)[36] on 17 October, 2022. N2 neuraminidase (NA) sequences of part of the associated H5 viruses in Taiwan were also downloaded from the same databases. The sequences in Taiwan were obtained through various sampling sources, primarily from cloacal/oropharyngeal swabs and tissues/droppings of suspected sick poultry in the farms[21,22,24], while some were taken from dead poultry in farms or rendering plants[22,26]. Short sequences (<1500 for HA and <1200 for NA) or sequences containing notable (n = 100) ambiguous nucleotides were removed from the download. Filtered genetic data were then aligned using Nextalign v2.3.0[37] or MAFFT v7.490[38] and trimmed to coding regions. Multiple basic amino acids at the H5 cleavage site were also removed. Sequences of Taiwan GsGd clade 2.3.4.4c viruses analysed in this study have all been described previously[21,22,24,26,39,40].

### Quantifying the persistence of GsGd lineages globally
The above GsGd H5 sequences (n = 15,576) were first subjected to location-focused downsampling. That is, for each country where genetic data were isolated, five sequences were randomly sampled on different dates within a monthly interval. When countries had samples with incomplete temporal information, up to 50 sequences were allowed to be randomly included per year. The downsampled data set (n = 5817) was then used to reconstruct a time-scaled tree using Treetime v0.11.1[41] and a maximum likelihood (ML) tree inferred by IQ-TREE v2.2.0[42], justified by a strong temporal signal ($R^2 = 0.95$). The *mugration* function in Treetime was performed to infer the country as ancestral states for each node. Post-introduction sublineages were defined by first identifying internal branches representing state transition and all the descending taxa of those branches. For each of these sublineages, nested transition branches were removed, resulting in a lineage each comprising taxa isolated in the same country and sharing a most common ancestor linked to viral movement. Duration of lineage persistence was defined by the maximum time interval of collection dates among the virus isolates assigned in a lineage. Considering the variation in sequencing effort across countries, an ancestral reconstruction based on transition rates informed by frequencies of HPAI H5 outbreaks during 2000–2022[43] was also performed using the *weights* function in Treetime.

### Phylogenetic analyses and data set design
A ML tree was inferred for the HA alignment using IQ-TREE. Major H5 viral lineages, including clades 2.3.4.4b and c, were classified according to the GsGd H5N1 nomenclature proposed by the World Health Organization[3,11]. All available H5 viruses isolated in Taiwan were identified on the tree. A total of 292 HA sequences isolated in Taiwan were classified as clade 2.3.4.4c, accounting for nearly all the available GsGd isolates in the country (292/295). The main working dataset was generated by removing duplicated sequences and viruses isolated in apparently the same outbreak (n = 252). H5N2 accounts for 62%, 59%, 75%, 100%, and 100% of the dataset from 2015 to 2019, whereas H5N8 accounts for 33%, 41%, 25%, 0%, and 0%. To verify results in the discrete phylogeographic analyses, more evenly distributed datasets were prepared by downsampling the sequences from Yunlin county (YL). The downsampled datasets, which include no more than six samples

per year in YL, contain comparable sample sizes as the other locations where the disease was prevalent (sample size in county YL, 26; CH, 27; CY, 24; PT, 26). In addition, to verify patterns in host diffusion, the N2 genes corresponding to the curated H5 datasets were collected (n = 183); N2 is the predominant subtype among clade 2.3.4.4c viruses in Taiwan (Supplementary Fig 3). Temporal signal in each data set was examined by TempEst v1.5.3[44], taking an ML tree inferred by IQ-TREE and times of sample collection as input. All phylogenetic trees in this study were visualised with the ggtree package[45].

### Epidemiological and spatial information of HPAI H5 viruses in Taiwan
The Taiwanese government has been actively conducting virological surveillance of HPAI H5 viruses in poultry farms and abandoned dead poultry prior to 2015[26,40]. Information on the collection date, host species and subtypes of each identification report with different sampling sources, including active surveillance or notification by owners/the public, can be found on a publicly available website[25]. The sampling sites' geographical coordinates for the reported events were obtained by parsing an interactive map embedded on the same website using the RSelenium package[46]. Metadata for the genetic data, including subtype, time, location at the county/city level and host, were acquired along with the sequences. Information on the sampling environment types, i.e., farms or slaughterhouses, was obtained from a recent publication by the Taiwanese government[22].

### Continuous phylogeographic analyses
We adopted a recently developed method to account for uncertainty in the sampling location for evolutionary reconstructions in continuous space[47,48]. Specifically, for each sequence, a prior describing multiple polygons and their corresponding probabilities were defined in a Keyhole Markup Language (KML) file and thereby incorporated into the Bayesian Markov chain Monte Carlo (MCMC) process performed by BEAST[49]. To construct the priors in this study, candidate locations for each genetic sequence were defined based on the sampling sites of the surveillance data. The candidate locations were gathered by matching the sequence's metadata, i.e. host, subtype, and county/city, to the reported events, along with a dynamic time-interval starting with (1) the collection date; if no hit was found with compatible information and in the time-interval, (2) the collection date ±7 was used. After conducting two searches, the centroids of all subdivisions in the county/city (i.e. town or area) of the sample were assigned to each unmatched sequence. The probabilities of a prior were uniform for sequences that were compatible with events in the surveillance data. However, for sequences that we were unable to link to reported events, the probabilities were assigned proportionally to the incidence of H5 HPAI during 2015–2019 in subdivisions. Each site with geographical coordinates was expanded to a minimum area of latitude ±0.0003 and longitude ±0.0005 to fit the polygon format in the original approach[47]. A sample collected in a slaughterhouse or rendering factory was assigned to the location of one arbitrarily selected slaughterhouse for each county/city based on the registry data (www.baphiq.gov.tw, accessed 22 March 2023).

Continuous phylogeographic analyses were performed with the Cauchy relaxed random walk (RRW) model implemented in BEAST v1.10.4[27,50]. A general time reversible (GTR) substitution model, a gamma-distributed rate, and an uncorrelated lognormal relaxed molecular clock[51] were employed in the Bayesian analyses, along with the Skygrid demographic model[52]. The MCMC was run for 550 million steps, with samples taken every 10,000 steps after removing 50 million steps as burn-in. Parallel runs were conducted to confirm the observed pattern. A starting tree generated from a simple BEAST calibration (identical clock parameters without traits) was added to each run to facilitate the sampling of tree topologies. The results of continuous phylogeographic analyses were visualised with the SERAPHIM package[53].

## Discrete phylogeographic analyses with an epoch model

To evaluate differences in spatial dispersal patterns throughout different epidemiological phases, from the introduction to the development of endemicity, we employed a time-heterogeneous transition rate model with the discrete phylogeographic method[54]. The method enables the Bayesian discrete state inference to accommodate multiple rate matrices, each of which is specified by one time interval (epoch). Therefore, transition frequencies and statistically supported directions in different intervals can be simultaneously estimated on the same phylogeny. We defined two time intervals, T1 and T2, divided by the date 1 September, 2016, based on sample sizes of the sequence data and the epidemiological trajectory (Fig. 2). Transition parameters were estimated using an asymmetric substitution model with Bayesian stochastic search variable selection (BSSVS)[55].

To better approximate the locations of the original farms, uncertain discrete state assignment was applied to the samples collected in slaughterhouses or rendering factories[56]. Similar to the strategy we used for the analyses in continuous space, we informed the priors of the uncertain discrete locations with surveillance data. We gathered candidate locations at the county/city level by matching the sequence's metadata to the reports within a 15-day interval (collection date ±7). If no matched events were found, we extended the interval to 90 days. The probabilities linked to the locations were proportional to the corresponding events in different counties and cities. The dispersal between hosts was also assessed by classifying sequences into the order of Anseriformes or Galliformes.

Additionally, we used a generalised linear model (GLM) incorporated into the discrete phylogenetic framework to investigate the potential predictors' contribution to the transition between locations[57]. The predictors included (1) road distance, (2) shared border, (3) number of poultry farms, (4) poultry population, (5) poultry heterogeneity, (6) area of cropland and (7) number of viral sequences. Poultry heterogeneity was calculated as Simpson's Index of Diversity[58]. Specifically, the value was calculated as $1-\Sigma(p_i)^2$, $p_i$ = proportion of the species (i.e. chicken, duck, goose or turkey) in the total poultry population. We combined the populations of chicken and duck as a single predictor because the two are highly correlated. Poultry farm, poultry heterogeneity and cropland area were added to the model as covariates for both origin and destination. Poultry farm, poultry population, poultry heterogeneity and sample size were treated as time-heterogeneous by averaging the annual values in the two intervals (2015–2016 and 2016–2019) (Supplementary Fig 2), while all predictors were log-transformed and standardised before inclusion in the matrices. The GLM model was adapted using previously described methods to assume temporally heterogeneous effect sizes and inclusion probabilities[13,59]. To account for unexplained variability, models that included time-homogeneous random effect variables specifying the effects as both origin and destination of each location were also implemented[60,61]. To validate the results, we performed reduced GLM models with no more than two predictors and a time-homogeneous GLM model including all the aforementioned predictors.

BEAST v1.10.5 (pre-released v0.1.2) was used to implement time-heterogeneous rate models with MCMC chains of 110 million steps[49]. Performance of the GLM estimations was improved by applying a set of empirical trees ($n$ = 1000) subsampled from a simple discrete-state BEAST run using LogCombiner[49]. All BEAST runs were facilitated by the BEAGLE library[62]. Convergence was examined using Tracer v1.7.2[63], ensuring that effective sampling sizes (ESS) were >200 for all continuous parameters.

Posterior analyses. The number of transitions between the two states was quantified by Markov jumps[64]. Bayes factors (BF) were calculated based on the indicators of transition rates resulting from BSSVS to determine statistically supported diffusion routes. Routes with BF > 20 were considered supported and were classified into two categories: 20–150 and >150, which are typically referred to as 'strong'

and 'very strong'[65]. The duration of viral persistence was evaluated using maximised time intervals that were unified by branch lengths, where both nodes were inferred to be in the same location. Trunk probabilities over time were summarised by PACT v0.9.4 (https://github.com/trvrb/PACT). Trunks were defined as ancestral branches shared by sequences isolated in 2019. Persistence and trunk probability were calculated using 1000 MCMC trees sampled from the posterior phylogeny distributions.

## Probing the spatial distribution of re-emergent outbreaks

Buffers representing disease hotspots on the map were created by parsing geographical coordinates inferred at the nodes of the posterior trees in the continuous phylogeographic calibrations, including tips and internal nodes. A buffer was defined as the area within 1 km of the inferred point location. The buffers created by the phylogenetic inference were merged with or without the buffers created by the sampling sites of contemporary outbreak events reported in the surveillance data for each tree. Only nodes estimated no earlier than 1 September 2016 (i.e. during the endemic phase, T2) and the outbreaks during the same period were taken as central points to create buffers. The nodes and the outbreaks were stratified by county/city before being transformed from points to polygons. Coverage ("captured") was determined by the sites of new outbreaks covered by the merged buffer areas. New outbreaks were defined as outbreaks occurring after the latest genetic sample, during 2019–2022. The captured outbreaks were summarised as proportions with 1000 posterior trees, whose node numbers were also used to generate randomly distributed null models. To evaluate the coverage performance, sensitivity analyses were conducted on buffer size considering multiple radius distances, including 0.5, 1 and 2 km. The spatial data manipulation was performed using the *sf* package[66]. The map data (shp files) were obtained from the government open data platform (data.gov.tw).

## Data sources of potential predictors

Road distances between counties and cities in Taiwan were measured with Google Maps. Data on agricultural statistics, including the number of poultry farms, poultry population and cropland area during 2015–2019 were obtained from the Agricultural Statistics Database of the Ministry of Agriculture (https://agrstat.moa.gov.tw/sdweb).

## Selection analyses

SLAC[67] and MEME[68] in the Datamonkey platform[69] were used to detect pervasive and episodic positive selection sites, respectively, with a default statistical significance level of $p < 0.1$. HyPhy v.2.3.14[70] was used to calculate non-synonymous/synonymous rate ratio, d$N$/d$S$ ($\omega$) estimates. The results of the surface proteins in the clade 2.3.4.4c viruses were compared with those in an enzootic, non-GsGd H5N2 lineage previously described in Taiwan[21,30].

## Reporting summary

Further information on research design is available in the Nature Portfolio Reporting Summary linked to this article.

# Data availability

The accession numbers of the Taiwan sequences analysed in our study are listed in Supplementary Table 2. The analysis results used to generate figures can be found in the project depository[71].

# Code availability

The custom code used in the analyses and the XML files required for BEAST can be found at https://github.com/yaotli/endemic[71].

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

## Acknowledgements

The work was supported by the UK Research and Innovation Global effort on COVID-19 (MR/V035444/1), Wellcome (207569/Z/17/Z, 224520/Z/21/Z), the UK Medical Research Council New Investigator Research Grant (MR/X002047/1) and a University of Glasgow Lord Kelvin/Adam Smith Fellowship to K.B. We are grateful for the continued surveillance efforts of the Animal and Plant Health Inspection Agency, MOA, Taiwan. We acknowledge the authors, originating laboratories, and submitting entities responsible for providing the sequences obtained from GISAID.

## Author contributions

Conceptualisation: Y.-T.L., H.-Y.K. Formal analysis: Y.-T.L. Methodology: Y.-T.L., J.H. Resources: H.-Y.K., Y.-L.L. Writing—original draft preparation: Y.-T.L. Writing—review & editing: Y.-T.L., H.-Y.K., J.H., M.-T.L., Y.-L.L., K.H., K.B. Supervision: J.H., K.H., K.B. Funding: K.H., K.B.

## Competing interests

The authors declare no competing interests.
