## [Transparent Peer Review file · Nature Communications]

From emergence to endemicity of highly pathogenic H5 avian influenza viruses in Taiwan

Corresponding Author: Dr Yao-Tsun Li

Version 0:

Reviewer comments:

Reviewer #1

(Remarks to the Author)

Main comments:

The authors present a timely, original and well-conducted phylogenetic analysis of the clade 2.3.4.4c avian influenza virus in Taiwan to address mechanisms of switch from emergence to endemicity. Utilizing state-of-the-art Bayesian phylogenetic inference, they highlight a methodology that, while not new, deserves wider application. They demonstrate that the virus has become endemic to the island through a comprehensive world-scale phylogeographic analysis, providing evidence that transmission has been sustained for several years within Taiwan. The study dates the transition from an initial epidemic to endemic circulation to 2016 and shows, through ancestral host reconstruction, that this shift coincides with a change in the primary host from mostly Anseriformes to mostly Galliformes.

Subsequently, the authors conducted a Taiwan-centered phylogeographic analysis to determine if local circulation is confined to a narrower area within the island. They performed both continuous and discrete phylogeographic analyses, using counties as spatial entities. The results indicate that endemicity is primarily sustained within a single county, with outbreaks in other counties largely following migration events, thus being sunk areas. This finding is crucial as it identifies the specific county where management measures should be concentrated.

Finally, the authors attempted to identify predictors that could explain the transmission routes. Although this part of the study did not find any significant predictor, it is noteworthy because it suggests that the proposed predictors, which all appeared relevant to test, are actually not critical in determining transmission routes.

A couple important elements are to clarify:

- it is not entirely explicit whether the endemic circulation stopped in 2023 – in which case the suggestion to focus management measures in Yunlan would be less relevant. Also not really clear why there is no sequence available since 2019 (because there are much fewer cases reported or because of a change in sequencing policy or sequence data sharing?)
- the whole sample collection process seems to be focused on carcasses, this needs to be made explicit.

And a couple of others would be important to discuss:

- The main limitation of this work is that it only includes domestic bird samples and not any wildlife sample. The authors probably can't address this limitation as no wildlife sample might be available, but they should discuss that limitation. The absence of any significant predictors of transmission routes is an additional reason to think wildlife could play a key role.
- All sequences used are from carcasses (or at least it seems to be the case, but this needs to be made explicit at stated above), which is a strong sampling bias when it comes to compare transmission rates between different hosts. There is a possibility that the virus actually evolved toward lower pathogenicity for Anseriformes and not for Galliformes. In that situation, most Anseriforme cases would have remained unnoticed and unsampled. Thus, the apparent switch from bidirectional transmission between Galliformes and Anseriformes to mostly unidirectional transmission from Galliformes to Anseriformes could just reflect the sampling bias. This scenario is not possible to exclude with this study design and thus needs to be discussed (and the authors should make it explicit that sampling of healthy bird is the only way to tackle that issue).

Other comments:

Abstract – make explicit whether the endemic circulation now stopped or not

L80 – Insert comma after ‘Since emergence’.

L88 – and at much larger scale among marine mammals in South America (Leguia, M., Garcia-Glaessner, A., Muñoz-Saavedra, B. et al. Highly pathogenic avian influenza A (H5N1) in marine mammals and seabirds in Peru. *Nat Commun* 14, 5489 (2023). <https://doi.org/10.1038/s41467-023-41182-0>)

L106 – Specify that those approximately 20% of outbreaks that occurred within the country were those reported for domestic birds ? (unnoticed outbreaks might have occurred in wild populations ?)

L121-L123 What has been done is not very clear, just timing based reconstruction of the spread between geographic occurrence? Or just a priori hypotheses based on that but that did not rely on any statistical inference?

Fig 1. and in the corresponding text section: it would be appropriate to define here how a lineage is defined (not only in the Methods)

L227 - ‘biased’, not ‘biased’

L 302-304 – The data analysis strategy is not mentioning the groups of host species (Galliformes, Anseriformes) until that part of the manuscript: wouldn’t it be better to inform the reader early on that the host species will eventually be considered? Were potential sampling biases considered in the analyses considering the host species? (as outlined earlier in the manuscript for the analyses not considering the host species, L218-225). More generally, it would be good to explain why the host species were not considered in the first series of analyses (e.g., by distinguishing farms with/without waterfowl; see L300-301).

L 316-319 – Please outline the assumption/hypothesis made when comparing the dN/dS ratio between the North American and Taiwan clades

L425 – Explicite the fact that the lineage persistence is the time between the later tip and the ancestral node.

L426 – 427 – 428 not clear what this second approach is

L433 – Unclear what “identified” means

L441 – Unclear why you need to use this N2 dataset and what you use it for

L446 – Make explicit if this surveillance is targeted on mortality events or if it’s random sampling of poultry farms, regardless of any a priori on the infectious status

L448 – What are ‘abandoned carcasses? Abandoned by whom and collected by whom? Is this expected to be random sampling outside/inside of poultry farms?

L481 – I assume the starting tree is the tree obtained at the previous step with Iqtree, make it explicit

L501 – So the underlying assumption is that the probability of an unreported event is negligible

L502-503 – It seems you might have enough data from the different host species to check that it makes sense to gather Galliformes together and Anseriformes together. It’s not trivial that viral adaptation matches this specific host phylogenetic levels, so it could be worth to check

L508 – Poultry heterogeneity refers to species heterogeneity? How is this calculated? Why not using the number of chicken and the number of ducks as different predictors? that could greatly improve the ability to identify differences between hosts

L512 – Standardise = normalised?

L521 – Unclear what ‘Performance of the GLM estimations were improved by applying a set of empirical trees’ means. How have those trees been obtained and what’s the rationale of using a set of trees rather than just the MCC? The tree topology is uncertain?

L557 – Move the definition of poultry heterogeneity to the line 508 where you first mention it.

END

Reviewer #2

(Remarks to the Author)

Reviewer #3

(Remarks to the Author)

This is a potentially interesting paper analysing the avian influenza H5Nx clade 2.3.4.4c outbreaks in Taiwan. Although these having occurred 2015-2019 pre-covid have been reported previously by several authors, it is interesting to revisit especially in the light of the recent (2024) 2.3.4.4b situation in the USA, and the 2014/2015 Eastern Asia - North America 2.3.4.4c outbreak connections.

The paper presents an interesting series of phylodynamic and evolutionary analyses, however I think there are several places where the text could be improved or expanded upon. Consequently I have some suggestions for clarifications and improvements for the manuscript:

Introduction

Line 74 (or similar) - I think that you need to describe more precisely what you mean by endemicity in the introduction - just citing reference 5 is not sufficient. For example, if you mean 'persistent circulation' - firstly how long must that persistence be ? (Suggest it should be more than 1 year at least); secondly if circulation - then that would imply initial import(s) to a defined geographic region (maybe the whole of Taiwan) and subsequent spread / circulation, and not ongoing multiple imports (re-seedings) over the time period; thirdly - in what population(s) ? suggest you are meaning resident birds - domestic and wild ?

Line 90 - [comment only] "establishment of endemic circulation"; the use of reference 16 is OK, but it is specifically about 2.3.4.4b in Europe and it is not clear that it would necessarily translate to Taiwan / Asia (the birds, situation, and reassortants are different).

Line 94 (and generally 92-101) - I think you need to expand upon the evolutionary history in Taiwan in the introductory text; I think references 18-20 are referring to the initial H5N8 introductions, and then it was later that H5N2 was circulating in Taiwan. So that is presumably at least 2 different introductions ? (And then note earlier comments about persistence and continuous circulation in the definition of endemic).

[note - probably not required for Introduction but the NA-N2 of the H5N2 in Taiwan is different to the NA-N2 of the H5N2 in North America (2015) ?; it is the HA-H5 (and maybe some other segments) which connect Taiwan to North America in 2015 ?]

Results

Lines 115-119 - please explain what you mean by sublineages; do you mean monophyletic clade within one country ? How big was the dataset (presumably more than 400 sequences..), and what does uniform downsampling mean ? What was the criterion for a sub lineage (posterior probability or bootstrap support at node, min number of sequences, min percentage of one country ?) - fine if this detail is in the methods but you should then reference in the text where these details are.

Line 134-141 - Taiwan; please also reference the pale blue sequences in figure 1.

Line 135 - what subtypes initially (2015) and subsequently ? (2016-2017). Please mention how big the sequence data set it (even if described in methods); and the proportion of subtypes per year / wave. [what I'm getting at here is how much of the first wave is H5N8 and how much of the 2nd is H5N2].

Line 158 - dispersal dynamics; I think you need to first establish in the main text what is monophyletic in HA-H5 (e.g. figure 1) before this part. Is the 2015/2016 one monophyletic clade in Taiwan (and not mixed with other countries), and 2016+ a new introduction to Taiwan but subsequently monophyletic (and not mixed with other countries) ?

Line 178 - please clarify what you mean here about the locations of the slaughterhouses and rendering factories; firstly are you talking about the precise locations (continuous values of latitude and longitude), or are you talking about the county in which the slaughterhouses are located ? Secondly, how much (e.g. percentage) of the sequence data do the slaughterhouses represent as opposed to premises with precise locations in each time period ? Thirdly, I think you are now trying to infer the location of the farms which sent their poultry to the slaughterhouses by correlating with outbreak reports or surveillance informed prior probabilities. But is it true that only one county would be sending poultry to a particular slaughterhouse in a time period (e.g. week); could the infected birds be coming from many dispersed individual farms ? If there are only a small number of slaughterhouse sequences compared to all, then this approach might work, but it is unclear the extent to which this re-assignment procedure has affected your subsequent results (see later). Therefore, you probably would want to omit these sequences altogether from the trees, or if performing a discrete traits phylogeography in BEAST, code these sequences with uncertain locations - in fact I think some of these details might be explained some more starting on line 457 ? Suggest putting in a reference to your methods section here anyway to clarify.

Line 188 - 190 - comment on 'statistically supported routes' - I think what you have done is probably sound enough but it is difficult to work out what effect the inference of the slaughterhouse and rendering factories locations have had on these results from the text.

Line 218 - Excellent that you have tried to downsample the data to test for robustness. As indicated above, it might be worth also trying this but without using the inferred locations of the farms linked to the slaughterhouse samples.

Line 227 - 229 - here you have used an uninformative prior for the unknown samples locations. Please can you indicate what percentage of all the samples (in the 2 different epochs) these were.

Line 231 - "reveals a more widespread distribution of dispersal origins.." - the problem is that this may well be true (in reality), but if you have used an uninformative prior on unknown sample locations then you might have artificially created a more widespread distribution yourself ? It is not clear from the text or main figures how many, where and in what time frames these unknown samples locations occur (can you include in a figure somewhere ? Maybe on figure 3 ?)

Line 273 - "re-emerging HPAI outbreaks" - please clarify what you mean; are these definitely re-emerging from directly from older outbreaks in Taiwan, are they new introductions from other countries, are they new reassortants ?

Line 300 - "The initial 2015 epidemic wave" - this was H5N8 ? and associated with wild migrating anseriformes in other countries ?

Line 307 etc - the sampling has likely influenced these exact results, however I think that supplementary figure 3 is

informative for this section, and suggest that you move it into the main text. It also shows the subtypes and indicates the number of slaughter houses etc with inferred locations. Perhaps combine supplementary figure 3 with figure 7 to a multi panel figure ?

Line 316 - 324 - good that you have included the dN/dS analyses and noted that there was not much (if any) signature of host selection. Line 322 'North American lineage' - do you mean the lineage mentioned in line 318 (2003) ? And did you also compare to the 2015 H5N2 in North America, which is also HPAI 2.3.4.4 ? (But the N2 are different ?). In terms of host adaptation signatures, (line 324), you might want to mention if Taiwan H5N2 still has the original wild bird associated 'long stalk' NA, and did not undergo a deletion to 'short stalk' NA which has previously been observed in relation to chicken adaptation in H5N1 and H9N2 ? [i.e. if you had seen a change to short stalk NA, then that would be a signature of adaptation to chickens but you did not see that either].

Discussion

Line 355 - "GLM did not identify any significant agricultural predictors" - OK but please also discuss the difference between multiple new incursions into Taiwan in 2016+ (which were reported as new reassortants) vs persistence within Taiwan.

Line 365-380 - please also discuss the effect of subtype (and other segments). Is it possible that the original H5N8 was well adapted to anseriformes, but the H5N2 was more adapted to galliformes especially considering the NA subtype (and where the N2 originated from, was it part of the H9N2 endemic constellations in Asia ?) and possibly other internal segment constellations ? And does that (or could that) partly explain the pattern ?

Methods

Line 457 & 496 - some of these details might be needed to be explained more in the main text (Results) please see above.

Line 522 - please be more explicit about the "empirical trees" - e.g. how many posterior trees (from what runs) were re-used ?

Line 557 - Poultry heterogeneity (and ref 65); what does this mean ? is it essentially entropy of anseriformes vs galliformes ?

Version 1:

Reviewer comments:

Reviewer #1

(Remarks to the Author)

The authors have taken into account of the comments made on the earlier version and the manuscript was revised appropriately. It is reporting important results on the transmission dynamics of high pathogenicity avian influenza virus during transition to endemicity in an asian country. It will be of interest to a broad audience.

Reviewer #2

(Remarks to the Author)

Reviewer #3

(Remarks to the Author)

Many thanks for considering my questions, and for your responses and changes to the manuscript. I think the changes have improved the clarity and answered my questions.

Just a very minor point - line 292 which now reads "Note that the re-emerging outbreaks here may include HPAI of different H5 lineages, without available genomic data for further investigation." should probably be "Note that the re-emerging outbreaks here may include HPAI of different H5 clade 2.3.4.4c reassortants, without available genomic data for further investigation." ? [because the lineages are defined according to the HA, and you've already shown that it is all 2.3.4.4c]

Reviewer #1 (Remarks to the Author):

Main comments:

The authors present a timely, original and well-conducted phylogenetic analysis of the clade 2.3.4.4c avian influenza virus in Taiwan to address mechanisms of switch from emergence to endemicity. Utilizing state-of-the-art Bayesian phylogenetic inference, they highlight a methodology that, while not new, deserves wider application. They demonstrate that the virus has become endemic to the island through a comprehensive world-scale phylogeographic analysis, providing evidence that transmission has been sustained for several years within Taiwan. The study dates the transition from an initial epidemic to endemic circulation to 2016 and shows, through ancestral host reconstruction, that this shift coincides with a change in the primary host from mostly Anseriformes to mostly Galliformes.

Subsequently, the authors conducted a Taiwan-centered phylogeographic analysis to determine if local circulation is confined to a narrower area within the island. They performed both continuous and discrete phylogeographic analyses, using counties as spatial entities. The results indicate that endemicity is primarily sustained within a single county, with outbreaks in other counties largely following migration events, thus being sunk areas. This finding is crucial as it identifies the specific county where management measures should be concentrated.

Finally, the authors attempted to identify predictors that could explain the transmission routes. Although this part of the study did not find any significant predictor, it is noteworthy because it suggests that the proposed predictors, which all appeared relevant to test, are actually not critical in determining transmission routes.

A couple important elements are to clarify:

- it is not entirely explicit whether the endemic circulation stopped in 2023 – in which case the suggestion to focus management measures in Yunlan would be less relevant. Also not really clear why there is no sequence available since 2019 (because there are much fewer cases reported or because of a change in sequencing policy or sequence data sharing?)

[Response]

H5 highly pathogenic avian influenza viruses continue to circulate in Taiwan. Based on the government-run online dashboard (ref. 25), which is also the primary epidemiological data source in our study, there have been 83 identification reports, in which Yunlin accounts for most of them (32), followed by Changhua (14) from January 2023 to July 2024. A screenshot of the website (<https://twai.baphiq.gov.tw/AI>) is shown below. We addressed the point in lines 367-369.

The agricultural authorities in Taiwan have distinct rules in data sharing, and in the country sample collection related to avian influenza has been regulated. The sequences isolated in 2019

were generated in a paper published in 2020 (Li 2020 [21]). Prior to this publication, only two sequences isolated after 2015 had been released to the public (see Figure 1 in that study, <https://academic.oup.com/ve/article/6/1/veaa037/5831843>). A large proportion of the genetic data analyzed in the current study were published by Huang et al., 2021 [22]. The study published in 2021, led by scientists in the agricultural departments in Taiwan, focused only on viruses circulating during 2015-2018 without releasing information post-2018.

[REDACTED]

- the whole sample collection process seems to be focused on carcasses, this needs to be made explicit.

[Response]

Sampling was conducted from different sources in the previous studies intended for virus detection. Based on the studies which performed the sample collection and generated the genetic data, most samples were taken by swabbing suspected sick animals during active surveillance (Huang 2016 [24], Huang 2021 [22]), so except for the specimens from rendering plants (2/295), and from Lee 2016 [26] (18/295), we cannot assume most were taken from dead birds. We have added details on sample sources in lines 425-428 for clarification.

And a couple of others would be important to discuss:

- The main limitation of this work is that it only includes domestic bird samples and not any wildlife sample. The authors probably can't address this limitation as no wildlife sample might be available, but they should discuss that limitation. The absence of any significant predictors of transmission routes is an additional reason to think wildlife could play a key role.

[Response]

We addressed this by adding lines 409-411 in the discussion.

- All sequences used are from carcasses (or at least it seems to be the case, but this needs to be made explicit as stated above), which is a strong sampling bias when it comes to compare transmission rates between different hosts. There is a possibility that the virus actually evolved toward lower pathogenicity for Anseriformes and not for Galliformes. In that situation, most Anseriforme cases would have remained unnoticed and unsampled. Thus, the apparent switch from bidirectional transmission between Galliformes and Anseriformes to mostly unidirectional transmission from Galliformes to Anseriformes could just reflect the sampling bias. This scenario is not possible to exclude with this study design and thus needs to be discussed (and the authors should make it explicit that sampling of healthy bird is the only way to tackle that issue).

[Response]

The genetic sequences analyzed in our study were primarily sampled from suspected sick birds, with some isolated from dead birds as indicated above and now detailed in lines 425-428. Indeed, our analyses assume that there has been little change in sampling strategies, especially referring to the active surveillance operated by the agriculture departments. We added lines 388-389 to address the issue. However, we would like to point out that the inference on virus dispersal did not entirely rely on the number of samples, and during 2016-2018 Anseriformes species samples still account for at least 1/4 of yearly isolates (Figure 7B). In addition, while evolution toward weakened pathogenicity in Anseriformes species is not impossible (as discussed in lines 377-378), viruses isolated in 2019 still maintained the molecular marker of highly pathogenic avian influenza (PLRERRRKR/GLF) identical to the isolates in 2015 (also stated in Li 2020 [21]). It is therefore expected that the viruses would remain virulent regardless of the bird species.

Other comments:

Abstract – make explicit whether the endemic circulation now stopped or not

[Response]

H5 highly pathogenic avian influenza viruses continue to circulate in Taiwan as answered above. However, whether GsGd clade 2.3.4.4c remains in circulation is unclear. The last official report on the isolation of clade 2.3.4.4c viruses was in January 2023 (in Chinese: <https://shorturl.at/ZoERe>), reported in line 357. We thus modified the sentence in the abstract “GsGd endemicity” to “GsGd endemicity up until 2023” for clarification that we focused only on the 2.3.4.4c although other GsGd lineages may also fit the model.

L80 – Insert comma after ‘Since emergence’.

[Response]

Done.

L88 – and at much larger scale among marine mammals in South America (Leguia, M., Garcia-Glaessner, A., Muñoz-Saavedra, B. et al. Highly pathogenic avian influenza A (H5N1) in marine mammals and seabirds in Peru. *Nat Commun* 14, 5489 (2023). <https://doi.org/10.1038/s41467-023-41182-0>)

[Response]

Edited as suggested by the reviewer (lines 88-89).

L106 – Specify that those approximately 20% of outbreaks that occurred within the country were those reported for domestic birds ? (unnoticed outbreaks might have occurred in wild populations ?)

[Response]

Corrected the sentence as “20% of reported outbreaks in poultry” (line 111).

L121-L123 What has been done is not very clear, just timing based reconstruction of the spread between geographic occurrence? Or just a priori hypotheses based on that but that did not rely on any statistical inference?

[Response]

The inference was made using the same approach as the first estimate (line 123), but adjusting the discrete state transition rates based on the outbreak information acquired from FAO. It calculated the equilibrium probabilities with the outbreak data provided, using the *weights* argument in Treetime, rather than solely relying on the frequencies of genetic data. We supplemented the detail in Methods line 448-449.

Fig 1. and in the corresponding text section: it would be appropriate to define here how a lineage is defined (not only in the Methods)

[Response]

The sentences in lines 119-121 were extended as follows: “we identified descendent sublineages led by introductory events. We found about 400 sublineages by reconstructing dispersal history from a uniformly downsampled dataset using the H5 genes of GsGd viruses up to September 2022.”

L227 - ‘biased’, not ‘biassed’

[Response]

Corrected.

L 302-304 – The data analysis strategy is not mentioning the groups of host species (Galliformes, Anseriformes) until that part of the manuscript: wouldn't it be better to inform the reader early on that the host species will eventually be considered? Were potential sampling biases considered in the analyses considering the host species? (as outlined earlier in the manuscript for the analyses not considering the host species, L218-225). More generally, it would be good to explain why the host species were not considered in the first series of analyses (e.g., by distinguishing farms with/without waterfowl; see L300-301).

[Response]

The original lines 300-301 indicate the transmission was found in the domestic waterfowl population rather than indicating the presence of these birds as a risk factor; we had no detailed information on the prevalence of mixed farming. We moved the sentence to Introduction (lines 104-106), and added line 115 to inform the readers earlier as suggested by the reviewer.

Both spatial and ecological data were analyzed with caution to mitigate potential biases. Creating subsamples and executing other sensitivity analyses with spatial dataset are due to the large differences in sample size between samples from Yunlin (>90) and other counties/cities (all <30) (Figure 2D). For ecological inference, sample sizes of the two host groups were comparable, and their relative sizes from 2015-2019 did not change drastically, except for 2019 when only 2 sequences were available (Figure 7B). We thus used another strategy to confirm our finding in viral ecology; that is, replicating the analyses using NA genes as previous studies have also done.

In addition, compared with the spatial information, the host information was explicit based on strain name or metadata linked to the viral sequences, so there was no particular need to estimate the unknown state of some isolates.

Without doubt, host range/tropism is critical for influenza epidemiology. However, virus transmission between different hosts/species is less straightforward in the context of Taiwan outbreaks. Most areas in Taiwan are highly urbanized and there have been few, if any, highly pathogenic avian influenza isolated in resident wild birds; in fact there's no clade 2.3.4.4c genetic data available from resident wild birds in Taiwan. Therefore, while the between-host transmission provided another important layer of insights into the evolution and epidemiology of clade 2.3.4.4c in Taiwan, our manuscript first focuses on the geographical dispersal of the lineage.

L 316-319 – Please outline the assumption/hypothesis made when comparing the dN/dS ratio between the North American and Taiwan clades

[Response]

We added the assumption for the comparison in lines 329-330 (North American H5N2 has been strictly isolated in the terrestrial poultry before 2000, illustrating long-term circulation in Galliformes hosts).

L425 – Explicite the fact that the lineage persistence is the time between the later tip and the ancestral node.

[Response]

Duration of lineage persistence was defined by the maximum time interval of collection dates among the virus isolates in the sublineage (added details in lines 445-446). We did not use the time directly inferred by Treetime because the relatively simple time-scaling method implemented by Treetime and the need to downsample full data resulted in less stable and less precise local time estimates based on previous estimated tMRCAs from BEAST. Using directly inferred dates resulted in a moderate increase in median duration (from 0.6 to 0.8) and an increase in the number of lineages circulating for more than 3 years (from 15 to 23). The duration of Taiwan lineage circulation also increased from 4.21 to 5.43; the latter was apparently divergent from previous estimations (based on the introduction time no earlier than the 2014 August [21,22] and latest isolated in March 2019). We therefore implemented a relatively conservative approach.

L426 – 427 – 428 not clear what this second approach is

[Response]

The inference was made using the same approach as the first estimate (lines 120-122 or lines 440-441), but adjusting the discrete state transition rates based on the outbreak information acquired from FAO. It calculated the equilibrium probabilities with the outbreak data provided, using the *weights* argument in Treetime, rather than solely relying on the frequencies of genetic data. We supplemented the detail in Methods line 448-449.

L433 – Unclear what “identified” means

[Response]

It basically means to locate all Taiwanese isolates on the global H5 phylogenetic tree. The procedure then led to the identification of numbers of sequences in the clade 2.3.4.4c and other clades (the following line).

L441 – Unclear why you need to use this N2 dataset and what you use it for

[Response]

The NA(N2) data were used to confirm our findings in the HA-based virus diffusion between host groups (Supplementary figure 10). A similar strategy was undertaken in previous studies (e.g. Trovao et al., 2015, MBE; Zhang et al., 2023, MBE). We did not apply the same strategy in the spatial analyses because there were fewer N2 genes in the early epidemic phase (2015-2016) and more discrete categories in the spatial analyses (>10 city/county vs 2 host group).

L446 – Make explicit if this surveillance is targeted on mortality events or if it's random sampling of poultry farms, regardless of any a priori on the infectious status

[Response]

We believe that the epidemiology data here reflect all known highly pathogenic avian influenza detection events logged by the agricultural authorities in Taiwan, and thus the samples were acquired by different routes, including “active surveillance”, “enhanced surveillance”, “retrospective investigation”, “reported by the owner” and “reported by the public” (detailed in the Report_type column of the baphiq_website.tsv file in the project depository). We modified lines 472-473 to “Information on the collection date, host species and subtypes of each identification report with different sampling sources including active surveillance or notification by owners/the public can be found on a publicly available website.”

L448 – What are ‘abandoned carcasses? Abandoned by whom and collected by whom? Is this expected to be random sampling outside/inside of poultry farms?

[Response]

The epidemiology data here reflect all known highly pathogenic avian influenza detection events logged by the agricultural authorities in Taiwan. Therefore, the dead poultry were collected and diagnosed positive by the agricultural departments. These records were logged as species “other” in the baphiq_website.tsv file and were all with “reported by the public” type of report. We changed the carcasses to dead poultry for clarification in lines 152 and 471.

L481 – I assume the starting tree is the tree obtained at the previous step with Iqtree, make it explicit

[Response]

The starting tree was generated by a simple BEAST run with the same parameters (following studies for instance, Van Borm et al., EID, 2023). We added the detail in lines 505-506.

L501 – So the underlying assumption is that the probability of an unreported event is negligible

[Response]

We assume the detection rate is higher than the sequencing rate given an affected poultry farm, and we believe that the epidemiology data here reflect all known highly pathogenic avian influenza detection events logged by the agricultural authorities in Taiwan as mentioned previously.

L502-503 – It seems you might have enough data from the different host species to check that it makes sense to gather Galliformes together and Anseriformes together. It's not trivial that viral adaptation matches this specific host phylogenetic levels, so it could be worth to check

[Response]

We agree that biological variation at the species level may contribute to different transmission patterns. The classification was primarily based on the perspective of the analysis, where the Galliformes and Anseriformes groups have comparable sample sizes (142 vs 110), while chickens account for over half of the samples (n=131). In addition, as all of these species are poultry, we assume their mobility was driven by human activity, and therefore the sequence data in our study was classified by Galliformes/Anseriformes, as in previous phylodynamic studies (ref. 35 and 40).

L508 – Poultry heterogeneity refers to species heterogeneity? How is this calculated? Why not using the number of chicken and the number of ducks as different predictors? that could greatly improve the ability to identify differences between hosts

[Response]

The value was calculated as $1 - \sum(p_i)^2$, p_i = proportion of the species (i.e. chicken, duck, goose or turkey) in the total poultry population. We added the details on heterogeneity in lines 534-546.

We combined the populations of chicken and duck as a single predictor because the two are highly correlated, especially in the county with higher disease incidence as indicated in the figure below. To avoid collinearity between predictors and overparameterization, as discussed in lines 369-370, we decided to include poultry population as a single predictor. We added lines 536-537 to explain this.

L512 – Standardise = normalised?

[Response]

Standardisation (std) is the term implemented in BEAUti. It's equal to normalisation based on the generated XML files.

L521 – Unclear what ‘Performance of the GLM estimations were improved by applying a set of empirical trees’ means. How have those trees been obtained and what’s the rational of using a set of trees rather than just the MCC? The tree topology is uncertain?

[Response]

Using starting tree(s) or empirical trees are methods to accelerate convergence during the MCMC sampling, as implemented in recent studies (Lemey et al., Nature, 2021; McCrone et al., Nature, 2022; Candido et al., Science, 2020). With empirical trees, the tree topologies were still sampled during the MCMC process (not fixed) but without considering all the priors on substitution. The empirical set used in our study was obtained by subsampling 1000 posterior trees from a BEAST run using LogCombiner. Details were added in lines 550-551.

L557 – Move the definition of poultry heterogeneity to the line 508 where you first mention it.

[Response]

The definition of heterogeneity was moved to line 534.

END

Reviewer #2 (Remarks to the Author):

Reviewer #2 (Remarks on code availability):

Although there is no readme file nor dedicated instruction, all xml files for the BEAST analyses are provided and ready to use, and the R codes are well annotated and easy to read and understand. It is thus straightforward for anyone that is familiar with BEAST and R to reproduce the results of the authors.

[Response]

We added a readme file to better replicate the analyses.

Reviewer #3 (Remarks to the Author):

This is a potentially interesting paper analysing the avian influenza H5Nx clade 2.3.4.4c outbreaks in Taiwan. Although these having occurred 2015-2019 pre-covid have been reported previously by several authors, it is interesting to revisit especially in the light of the recent (2024) 2.3.4.4b situation in the USA, and the 2014/2015 Eastern Asia - North America 2.3.4.4c outbreak connections.

The paper presents an interesting series of phylodynamic and evolutionary analyses, however I think there are several places where the text could be improved or expanded upon. Consequently I have some suggestions for clarifications and improvements for the manuscript:

Introduction

Line 74 (or similar) - I think that you need to describe more precisely what you mean by endemicity in the introduction - just citing reference 5 is not sufficient. For example, if you mean 'persistent circulation' - firstly how long must that persistence be ? (Suggest it should be more than 1 year at least); secondly if circulation - then that would imply initial import(s) to a defined geographic region (maybe the whole of Taiwan) and subsequent spread / circulation, and not ongoing multiple imports (re-seedings) over the time period; thirdly - in what population(s) ? suggest you are meaning resident birds - domestic and wild ?

[Response]

We admittedly don't have a clear definition for the endemicity of GsGd, and thus this led to our global profile to quantitatively evaluate the duration of GsGd in Taiwan (Figure 1). In line 74,

these countries were classified by FAO, probably because human infections were also identified in countries along with the outbreaks in farms. We believe the list has not been updated since then. The endemic state in Taiwan was claimed by authors in reference 22. We modified the line 100-102 to “With multiple genotypes generated by reassortment with local low pathogenic avian influenza (LPAI) viruses, Huang et al., recognized the circulation of the HPAI clade 2.3.4.4c virus in Taiwan as endemic”. Based on available data (please see figure in the later response), the circulation of clade 2.3.4.4c in Taiwan was caused by a single introduction without evidence of re-seeding (explained in lines 102-103). Globally, the lineage was only detected in Taiwan after 2016-2017 (lines 98-99). In terms of host, 2.3.4.4c has only been isolated in poultry (addressed in lines 105-106).

In our global persistence analysis, we focused on each (sub)lineage by identifying a single corresponding introductory event, instead of focusing on the geographical areas (lines 119-122). We considered isolates from any host in the analysis. With the analysis, we found the circulation of 2.3.4.4c in Taiwan was significantly longer than most circulation lineages led by introduction events, and thus readily fit the term.

Line 90 - [comment only] “establishment of endemic circulation”; the use of reference 16 is OK, but it is specifically about 2.3.4.4b in Europe and it is not clear that it would necessarily translate to Taiwan / Asia (the birds, situation, and reassortants are different).

[Response]

The reference was cited not to interpret the situations in Asia, but to point out the concern of developing long-term circulation in Europe. Our study aims to share our insights in the development of long-term virus circulation, which is relevant to areas currently having difficulty eliminating GsGd 2.3.4.4b, including but not limited to Europe. We added lines 90-92 to address the issue.

Line 94 (and generally 92-101) - I think you need to expand upon the evolutionary history in Taiwan in the introductory text; I think references 18-20 are referring to the initial H5N8 introductions, and then it was later that H5N2 was circulating in Taiwan. So that is presumably at least 2 different introductions ? (And then note earlier comments about persistence and continuous circulation in the definition of endemic).

[note - probably not required for Introduction but the NA-N2 of the H5N2 in Taiwan is different to the NA-N2 of the H5N2 in North America (2015) ?; it is the HA-H5 (and maybe some other segments) which connect Taiwan to North America in 2015 ?]

[Response]

The clade 2.3.4.4c in Taiwan is most likely the result of a single introduction, or of multiple introductions of the same lineage/subtype of the virus occurring in a very short time window,

which from an evolutionary perspective is equivalent to a single introduction. This finding has been demonstrated by previous studies ([21-24]), and by the figure below using all sequence data available in October 2022. In panel A, the dark tips represent viruses isolated in Taiwan, whereas panel B illustrates the expanded section of the orange branches in panel A. To our knowledge no new GsGd genomic sequences have been published since.

The earliest 2.3.4.4c in Taiwan was probably the H5N8 subtype based on its closest virus isolated in Japan [21,24] and the subtype distribution shown in Supplementary figure 3. Its descendants, identified by HA, then acquired NA and internal genes from different local low-pathogenic avian influenza viruses in Taiwan (lines 100-101). The genetic components of the GsGd viruses in Taiwan, except for HA, therefore were distinct from the viruses isolated in North America after substantial circulation (Figure 2 and Supp. figure in Li 2020 [21], <https://academic.oup.com/ve/article/6/1/veaa037/5831843>). We decided not to focus on the genetic origins of the lineage in Taiwan because they have been clearly described in multiple studies ([21-24, 31]), and are not directly relevant to the epidemiological interpretations in the current study. The sentence in lines 99-100 was modified to clarify that the 2.3.4.4c lineage changed its NA component after introduction.

Results

Lines 115-119 - please explain what you mean by sublineages; do you mean monophyletic clade within one country ? How big was the dataset (presumably more than 400 sequences..), and what does uniform downsampling mean ? What was the criterion for a sub lineage (posterior probability or bootstrap support at node, min number of sequences, min percentage of one

country ?) - fine if this detail is in the methods but you should then reference in the text where these details are.

[Response]

Each sublineage was first defined by a single corresponding introductory event, followed by excluding nested events if any. The introductory events were identified on internal branches representing state transition (lines 441-443). Therefore, in the first step viruses in a sublineage do form a monophyletic group (sharing the same ancestor node inferred as a foreign virus), among which viruses migrating to other countries are then removed. We did not apply statistical support to identify sublineages, but used different strategies to conduct the ancestral reconstruction (*migration* only or *migration* plus *weights* by outbreak frequencies) that detected introductory events (lines 448-449). The aim here is to summarize the durations of persistence in all detectable lineages. We also modified the text in lines 119-121.

The tree in Figure 1 was inferred with 5817 viral HA sequences, including viruses in Taiwan, downsampled from all GsGd H5 sequences available in 17, October, 2022 (n=15,576) (line 435). The uniform here means the same amount of sequences were sampled in the same time interval per country, without considering the genetic diversity (details in lines 435-437). The term and the strategy was suggested by Hall et al., 2016, Virus evolution. We added details on sample size in line 435 and 438.

Line 134-141 - Taiwan; please also reference the pale blue sequences in figure 1.

[Response]

The primary data set for the dispersal analyses (Figure 3-7) was prepared from all available clade 2.3.4.4c H5 sequences excluding duplicated sequences and viruses isolated in apparently the same outbreak (described in lines 454-457). The Taiwanese taxa shown in Figure 1 was the result of downsampling together with other countries.

Line 135 - what subtypes initially (2015) and subsequently ? (2016-2017). Please mention how big the sequence data set it (even if described in methods); and the proportion of subtypes per year / wave. [what I'm getting at here is how much of the first wave is H5N8 and how much of the 2nd is H5N2].

[Response]

H5N2 is the yearly dominant subtype during 2015-2019, accounting for 57, 60, 79, 95, and 92 % of the total reported isolates. The proportions of reported isolates of H5N8 are 25, 9.5, 5.4, 1.0, 0 (%). The rest of isolates are untyped H5 (31% in 2016) or mixed subtypes. Raw data are available in the Subtype column of the baphiq_website.tsv file in the project depository. The subtype information of the genomic data is illustrated in Supplementary figure 3. Proportions

of H5N2 sequences from 2015-2019 are 62, 59, 75, 100, and 100%, whereas H5N8 are 33, 41, 25, 0, 0%. We added details on sample size and subtype in lines 144-145 and 457-458.

Line 158 - dispersal dynamics; I think you need to first establish in the main text what is monophyletic in HA-H5 (e.g. figure 1) before this part. Is the 2015/2016 one monophyletic clade in Taiwan (and not mixed with other countries), and 2016+ a new introduction to Taiwan but subsequently monophyletic (and not mixed with other countries) ?

[Response]

The clade 2.3.4.4c in Taiwan is most likely the result of a single introduction, or multiple introductions of the same lineage/subtype of virus occurring in a very short time window (answered previously). We agree it's important to clarify the issue and thus added the sentence "and the circulating viruses were all linked to the same introduction event occurring before 2015" in lines 102-103.

Line 178 - please clarify what you mean here about the locations of the slaughterhouses and rendering factories; firstly are you talking about the precise locations (continuous values of latitude and longitude), or are you talking about the county in which the slaughterhouses are located ? Secondly, how much (e.g. percentage) of the sequence data do the slaughter houses represent as opposed to premises with precise locations in each time period ? Thirdly, I think you are now trying to infer the location of the farms which sent their poultry to the slaughterhouses by correlating with outbreak reports or surveillance informed prior probabilities. But is it true that only one county would be sending poultry to a particular slaughterhouse in a time period (e.g. week); could the infected birds be coming from many dispersed individual farms ? If there are only a small number of slaughterhouse sequences compared to all, then this approach might work, but it is unclear the extent to which this reassignment procedure has affected your subsequent results (see later). Therefore, you probably would want to omit these sequences altogether from the trees, or if performing a discrete traits phylogeography in BEAST, code these sequences with uncertain locations - in fact I think some of these details might be explained some more starting on line 457 ? Suggest putting in a reference to your methods section here anyway to clarify.

[Response]

Here (line 182) we tried to explain the issue caused by assigning exact positions to samples collected in slaughterhouses (or rendering factors) in the previous continuous Bayesian spatial analyses. That is, viruses collected in slaughterhouses should not be the origin of transmission, especially those slaughterhouses in the urban areas. Therefore it is more appropriate to estimate them coming from farms. There are 40 sequences isolated in slaughterhouses. The way we dealt with samples isolated in slaughterhouses was in fact to apply uncertain state assignment as

described in lines 521-528. The uncertain assignment for continuous spatial analyses were described in lines 481-495.

The re-assignment of samples collected from slaughterhouses was performed by linking one sequence to farms in multiple countries/cities having compatible isolate information, so the model is isolate-oriented without considering samples per area/per farm.

Line 188 - 190 - comment on 'statistically supported routes' - I think what you have done is probably sound enough but it is difficult to work out what effect the inference of the slaughterhouse and rendering factories locations have had on these results from the text.

[Response]

This is a good point and is one of the reasons we also ran the analysis without considering the sampling environment (starting in line 232, Supplementary Figure 4D, 5D, 6D). The location was assigned as its collection location, except for samples without any information on location that were assigned an uninformative prior.

Line 218 - Excellent that you have tried to downsample the data to test for robustness. As indicated above, it might be worth also trying this but without using the inferred locations of the farms linked to the slaughterhouse samples.

[Response]

We ran the analysis without considering the sampling environment (starting in line 232, Supplementary Figure 4D, 5D, 6D). The location was assigned as its collecting location, except for samples without any information on location that were assigned an uninformative prior.

Line 227 - 229 - here you have used an uninformative prior for the unknown samples locations. Please can you indicate what percentage of all the samples (in the 2 different epochs) these were.

[Response]

The detailed spatial information is illustrated in Supplementary Figure 3. There are 11 samples without any information on location (county/city); 10 were isolated in the first epoch and one was isolated in the second epoch. We supplemented the details in line 234.

Line 231 - "reveals a more widespread distribution of dispersal origins.." - the problem is that this may well be true (in reality), but if you have used an uninformative prior on unknown sample locations then you might have artificially created a more widespread distribution yourself ? It is not clear from the text or main figures how many, where and in what time frames these unknown samples locations occur (can you include in a figure somewhere ? Maybe on figure 3 ?)

[Response]

We considered each sensitivity analysis with the same weight, and they collectively present a similar dispersal model. The model shows that in the first epidemic phase viruses originated from different areas, and in the second phase basically all viruses originated from one county. As the data here referred to the supported dispersal routes, which should not be easily determined by samples with unknown spatial state (n=11). There are 11 samples without any information on location (county/city); 10 were isolated in the first epoch and one was isolated in the second epoch. The detailed spatial information is illustrated in Supplementary Figure 3.

Line 273 - “re-emerging HPAI outbreaks” - please clarify what you mean; are these definitely re-emerging from directly from older outbreaks in Taiwan, are they new introductions from other countries, are they new reassortants ?

[Response]

Re-emerging indicates the outbreak occurred after previous identification of viruses in the same geographical area, as indicated in line 297 and 574-575. It's possible that the re-emerging outbreaks were led by other lineages of highly pathogenic influenza viruses, as discussed in lines 404-406. However, the analyses here were intended (1) to verify if our phylogenetic reconstruction is epidemiologically relevant using new (test) data; and (2) to verify the role of the endemic hotspot in Taiwan. Therefore, the analyses still served the purpose if the new outbreak data included other HPAI viruses, indicating a more general transmission pattern in Taiwan. However, based on the tree shown previously using all available genomic data, there has been little information on lineages of other viruses. We added lines 291-292 to further address the issue.

Line 300 - “The initial 2015 epidemic wave” - this was H5N8 ? and associated with wild migrating anseriformes in other countries ?

[Response]

The outbreaks in 2015 were caused by viruses of different subtypes, including H5N2 (n=541) and H5N8 (n=232). The earliest 2.3.4.4c in Taiwan was probably the H5N8 subtype based on its closest virus isolated in Japan [21,24]. We have no clue as to which species of bird spread the virus to Taiwan. It is plausible that a species of waterfowl carried the virus to Taiwan through a migratory pathway.

Line 307 etc - the sampling has likely influenced these exact results, however I think that supplementary figure 3 is informative for this section, and suggest that you move it into the main text. It also shows the subtypes and indicates the number of slaughter houses etc with inferred locations. Perhaps combine supplementary figure 3 with figure 7 to a multi panel figure ?

[Response]

Our analyses assume that there has been little change in sampling strategies, especially referring to the active surveillance operated by the agriculture departments. We added lines 388-389 to address the issue. We prefer keeping the figure (Supp. fig. 3) in the supplementary primarily considering the size limitation allowed in the main text (and it would take up almost the whole page if combined with figure 7), and the figure contains mixed results of analyses in different sections. We would be willing to combine the two if that's preferred and allowed by the editor.

Line 316 - 324 - good that you have included the dN/dS analyses and noted that there was not much (if any) signature of host selection. Line 322 'North American lineage' - do you mean the lineage mentioned in line 318 (2003) ? And did you also compare to the 2015 H5N2 in North America, which is also HPAI 2.3.4.4 ? (But the N2 are different ?). In terms of host adaptation signatures, (line 324), you might want to mention if Taiwan H5N2 still has the original wild bird associated 'long stalk' NA, and did not undergo a deletion to 'short stalk' NA which has previously been observed in relation to chicken adaptation in H5N1 and H9N2 ? [i.e. if you had seen a change to short stalk NA, then that would be a signature of adaptation to chickens but you did not see that either].

[Response]

The selection signatures of Taiwanese 2.3.4.4c viruses were compared with a H5N2 lineage that originated from Mexico (not belonging to GsGd, as indicated by the previous figure)(lines 323-324). We did not compare Taiwanese 2.3.4.4c viruses with 2015 North American 2.3.4.4c viruses, because the circulation in the US/North America was relatively short (<3 years).

The N2 of the North American lineage in Taiwan indeed had 20 amino acid deletions (also indicated in a previous study [31]), but N2 of the Taiwanese 2.3.4.4c viruses did not. We appreciate the reviewer's comment and added this point to the discussion in lines 386-388.

Discussion

Line 355 - “GLM did not identify any significant agricultural predictors” - OK but please also discuss the difference between multiple new incursions into Taiwan in 2016+ (which were reported as new reassortants) vs persistence within Taiwan.

[Response]

The clade 2.3.4.4c in Taiwan is most likely the result of a single introduction, or by multiple introductions of the same lineage/subtype of virus occurring in a very short time window (answered previously). We agree it’s important to clarify the issue and thus added the sentence “and the circulating viruses were all linked to the same introduction event occurring before 2015” in lines 102-103.

Line 365-380 - please also discuss the effect of subtype (and other segments). Is it possible that the original H5N8 was well adapted to anseriformes, but the H5N2 was more adapted to galliformes especially considering the NA subtype (and where the N2 originated from, was it part of the H9N2 endemic constellations in Asia ?) and possibly other internal segment constellations ? And does that (or could that) partly explain the pattern ?

[Response]

A simple contingency table laying subtype (N2/N8) against host (Anseriformes/Galliformes) shows no support for correlation ($p=0.74$). The N2 of the 2.3.4.4c viruses in Taiwan originated from the gene pool of low pathogenic avian influenza that was not directly linked to H9N2 (Li 2020 [21]). We agree that the effect of internal genes could not be evaluated in our study, and added lines 408-411 to address this limitation.

	N2	N8
Anseriformes	81	25
Galliformes	104	37

Methods

Line 457 & 496 - some of these details might be needed to be explained more in the main text (Results) please see above.

[Response]

We explicitly mentioned the methods accounting for uncertainty in line 166 and 188.

Line 522 - please be more explicit about the "empirical trees" - e.g. how many posterior trees (from what runs) were re-used ?

[Response]

The empirical set used in our study was obtained by subsampling 1000 posterior trees from a BEAST run using LogCombiner. The details were added in lines 550-551.

Line 557 - Poultry heterogeneity (and ref 65); what does this mean ? is it essentially entropy of anseriformes vs galliformes ?

[Response]

The value was calculated as $1 - \sum(p_i)^2$, p_i = proportion of the species (i.e. chicken, duck, goose or turkey) in the total poultry population. We added the details on heterogeneity in lines 534-536.

Reviewer #3 (Remarks on code availability):

I have not attempted to run the code, however I have accessed and read the scripts which seem OK.

Reviewer #1 (Remarks to the Author):

The authors have taken into account of the comments made on the earlier version and the manuscript was revised appropriately. It is reporting important results on the transmission dynamics of high pathogenicity avian influenza virus during transition to endemicity in an Asian country. It will be of interest to a broad audience.

Reviewer #2 (Remarks to the Author):

Reviewer #2 (Remarks on code availability):

The authors added a useful README to quickly be able to reproduce the figures.

Reviewer #3 (Remarks to the Author):

Many thanks for considering my questions, and for your responses and changes to the manuscript. I think the changes have improved the clarity and answered my questions.

Just a very minor point - line 292 which now reads "Note that the re-emerging outbreaks here may include HPAI of different H5 lineages, without available genomic data for further investigation." should probably be "Note that the re-emerging outbreaks here may include HPAI of different H5 clade 2.3.4.4c reassortants, without available genomic data for further investigation." ? [because the lineages are defined according to the HA, and you've already shown that it is all 2.3.4.4c]

[Response]

We would like to thank all the reviewers for their thoughtful comments.

It's true that the hotspots were inferred by the HA genes of the 2.3.4.4c viruses collected up until 2019. However, we do not have genomic information (including HA) for the re-emerging outbreaks (2019-2022), but know that they are recorded as HPAI H5. We agree with the reviewer that most of these should be 2.3.4.4c reassortants and have revised the sentence to "Note that the re-emerging outbreaks here may include HPAI of different H5 lineages such as clade 2.3.4.4c reassortants, without ...".